# Analyzing the Thermal Characteristics of Three Lining Materials for Plantar Orthotics

**DOI:** 10.3390/s24092928

**Published:** 2024-05-04

**Authors:** Esther Querol-Martínez, Artur Crespo-Martínez, Álvaro Gómez-Carrión, Juan Francisco Morán-Cortés, Alfonso Martínez-Nova, Raquel Sánchez-Rodríguez

**Affiliations:** 1Clinic Sciences Department, Medicine and Health Sciences Faculty, University of Barcelona, 08080 Barcelona, Spain; equerolm@ub.edu (E.Q.-M.);; 2Nursing Department, Medicine and Health Sciences Faculty, Universidad Complutense de Madrid, 28080 Madrid, Spain; alvgom25@ucm.es; 3University Centre of Plasencia, Nursing Department, Universidad de Extremadura, 10600 Plasencia, Spain; juanfmoran@unex.es (J.F.M.-C.); rsanrod@unex.es (R.S.-R.)

**Keywords:** thermography, lining materials, orthoses, insoles, wearables

## Abstract

Introduction: The choice of materials for covering plantar orthoses or wearable insoles is often based on their hardness, breathability, and moisture absorption capacity, although more due to professional preference than clear scientific criteria. An analysis of the thermal response to the use of these materials would provide information about their behavior; hence, the objective of this study was to assess the temperature of three lining materials with different characteristics. Materials and Methods: The temperature of three materials for covering plantar orthoses was analyzed in a sample of 36 subjects (15 men and 21 women, aged 24.6 ± 8.2 years, mass 67.1 ± 13.6 kg, and height 1.7 ± 0.09 m). Temperature was measured before and after 3 h of use in clinical activities, using a polyethylene foam copolymer (PE), ethylene vinyl acetate (EVA), and PE-EVA copolymer foam insole with the use of a FLIR E60BX thermal camera. Results: In the PE copolymer (material 1), temperature increases between 1.07 and 1.85 °C were found after activity, with these differences being statistically significant in all regions of interest (*p* < 0.001), except for the first toe (0.36 °C, *p* = 0.170). In the EVA foam (material 2) and the expansive foam of the PE-EVA copolymer (material 3), the temperatures were also significantly higher in all analyzed areas (*p* < 0.001), ranging between 1.49 and 2.73 °C for EVA and 0.58 and 2.16 °C for PE-EVA. The PE copolymer experienced lower overall overheating, and the area of the fifth metatarsal head underwent the greatest temperature increase, regardless of the material analyzed. Conclusions: PE foam lining materials, with lower density or an open-cell structure, would be preferred for controlling temperature rise in the lining/footbed interface and providing better thermal comfort for users. The area of the first toe was found to be the least overheated, while the fifth metatarsal head increased the most in temperature. This should be considered in the design of new wearables to avoid excessive temperatures due to the lining materials.

## 1. Introduction

The temperature on the plantar surface of the foot is an important factor in comfort during daily life activities, especially in relation to interaction with socks, stockings, or footwear. This can be affected by different thermal properties such as friction with these elements, thermal diffusivity or conductivity, and mechanical properties. Furthermore, in patients with certain foot pathologies, orthopedic treatment is added, performed through the combination of various materials (thermoplastics, resins, and foams) that may interfere with proper thermoregulation of the plantar surface, causing excess heat and moisture, discomfort [1], or even the onset of dermal pathologies derived from sweating [2].

Materials used in foot orthoses or shoe insoles that come into direct contact with the foot must offer a quality bonus (cushioning, permanent elasticity, and stabilization), hygiene (disinfection), safety, traceability, and verified compatibility with human skin, so as not to cause toxic or irritating intolerances. However, this does not always positively affect perspiration, as they tend to retain heat. Thermal comfort in a foot orthosis is important for user acceptance; therefore, materials that generate excessive heat will not be as well tolerated as those that are more thermoregulatory. Performing a thermal analysis of these insole covering materials could provide quantitative information about the heating of such materials during use. Thus, a material that better adapts to the patient’s characteristics, is more comfortable, and generates greater treatment adherence could be chosen. 

The inclusion of novel materials (textiles with a 3D structure) has sought to improve thermal comfort through their porosity and breathability. Although no significant changes in foot plantar temperature have been found compared to traditional materials (leather), a reduction of up to 24.41% in heel moisture has been observed [3]. Other materials such as ethylene vinyl acetate (EVA), polyethylene foam (PE), or polyethylene–EVA (PE-EVA) did not show a negative impact on foot plantar temperature increase after a 3-h uninterrupted use session [4]. However, certain characteristics such as the color of polyethylene foam (PE) or materials with high friction coefficients should be avoided in the heel area to prevent overheating [4]. Nonetheless, the impact on the material’s own reheating is unknown, which in prolonged use sessions could cause discomfort, in addition to influencing its possible mechanical characteristics. This is also important in the design of wearables in the form of pressure and/or temperature insoles because the choice of lining material could bias the results [5]. Despite the importance of this fact, which would allow for better adaptation of treatments according to the characteristics of the patient, this topic has received scant attention in the literature.

The working hypothesis is that some intrinsic characteristics, such as the type of material, density, or coefficient of friction, may have an impact on material heating, which could negatively affect comfort perception. Therefore, the aim of this study was to assess the temperature increase reflected in three types of foams commonly used as lining for foot orthoses after a 3-h period of use in a clinical setting.

## 2. Materials and Methods

A total of 36 subjects participated in the study, comprising 21 women and 15 men, with a mean age of 24.6 ± 8.2 years, a mean height of 1.7 ± 0.09 m, and a mean mass of 67.1 ± 13.6 kg (Table 1). Anthropometric characteristics according to gender are presented in Table 2. All participants provided informed consent, adhering to the principles and guidelines of the Declaration of Helsinki. The study was approved by the Bioethics and Biosafety Committee of the University of Extremadura under registration number 186/2020. 

### 2.1. Study Protocol

To ensure that thermal measurements of body temperature were not altered, the participants were instructed not to engage in intense physical exercise during the 24 h prior to the start of the study. They were also advised to avoid consuming heavy meals, taking medication, applying cosmetic products to the skin, or consuming stimulants such as alcohol, tobacco, coffee, or tea in the 12 h before the start of the study [6,7].

Three different days were used to perform thermographic measurements, one for each material. For analysis, a pair of insoles made of each material were used, labeled with the participant’s shoe size and marked with the participant’s identifier. Below are the technical characteristics of the three materials used:Sidas-Podiatech. Podialene 125. 3 mm thickness, non-perforated. Expanded polyethylene copolymer (PE) foam with an open-cell structure. Hardness: approximately 25 Shore A. Density: approximately 0.11 g/cm^3^. Medium friction coefficient. Color: red.Nora^®^ Lunatur 27 Walnut. 3 mm thickness, non-perforated. Ethylene vinyl acetate (EVA) with a closed-cell structure. Hardness: approximately 27 Shore A. Density: approximately 0.24 g/cm^3^. Low friction coefficient. Color: light brown.Sidas-Podiatech. Podiamic 160. 3 mm thickness, non-perforated. Expanded polyethylene–ethylene vinyl acetate (PE-EVA) copolymer foam with an open-cell structure. Hardness: approximately 35 Shore A. Density: approximately 0.145 g/cm^3^. Low friction coefficient. Color: skin tone.

The assignment of materials to subjects was conducted through a random procedure, ensuring that each participant used a different material on consecutive days. Thus, the participant initially assigned material 1 (PE) was provided with material 2 (EVA) on the second day and material 3 (PE-EVA) on the third. This methodology allowed for the evaluation of all three materials on the same day but in different subjects. To maintain participant blinding regarding the evaluation order of the materials, a researcher was responsible for inserting and removing them from the footwear as needed, thus avoiding any potential bias in the subjects’ perception of the evaluated materials. Environmental conditions of 18–20 °C and relative humidity of 40–45% were established (controlled using a Flir MR77 thermo-hygrometer) to ensure that thermographic measurements were not affected by these factors and provide accurate readings [8].

The same protocol was established for all sessions. The researcher placed the insoles on a black thermal screen to prevent heat from being diverted by surrounding people or objects. Photos were captured in an area devoid of reflective lights, with uniform lighting, and the camera was precisely aligned with the appropriate lens focus and visual settings adjusted. Thermal snapshots of the materials were obtained using a FLIR E60BX thermal camera, with the following specifications: (1) a pixel count of 76.800, (2) thermal sensitivity of 0.045 °C at 30 °C, (3) a temperature oscillation of −20 °C to 120 °C with an accuracy of ±2% or 2 °C, and (4) a spectral range of 7.5 μm to 13 μm. The camera was placed on a tripod one meter away from the black screen and the material. Three thermographic images (emissivity of 0.98; iron bow color palette) were taken before inserting it into the participant’s usual medical footwear (same brand and model: Medical Shoes Zale^®^, Alicante, Spain).

To reduce the post-exercise thermal readjustment time, the subjects did not wear socks or stockings. Subsequently, over a period of three hours, the study participants engaged in their routine caregiving activities, which included intervals of walking, moments of static standing, and periods of sitting. These activities reflect the typical dynamics of a workday in a clinical environment, carried out in rooms with homogeneous surfaces and no slopes. After three hours, the participants returned to the study room, and the researcher removed the insoles from the footwear and positioned them again on the black surface to acquire three thermographic images.

The thermographic images were processed using Flir Tools v6.4 software, establishing 6 regions of interest (ROIs) on the insole material that correspond anatomically to the plantar surface of the foot. These regions included the center of the heel, the midpoint of the outer arch (styloid process), the first metatarsal head, the third metatarsal head, the fifth metatarsal head, and the first toe (see Figure 1).

### 2.2. Statistical Analysis

To maintain data independence [9], all variables analyzed pertain to the left insole, which was randomly selected. The mean temperature was calculated for each zone of the images captured for each lining material before and after physical exercise. After confirming that the sample data adhered to normality (Kolmogorov–Smirnov test, *p* > 0.05 in all cases), the following analyses were conducted: (1) a paired-sample *t*-test to assess pre-post temperature differences and (2) Pearson correlations to identify if the room temperature influenced the subjects’ plantar temperature. Additionally, since the temperature data also met the assumption of sphericity (*p* > 0.05 in all 3-layer comparisons), repeated-measures ANOVA (3 × 3, with Bonferroni confidence model adjustment) was performed to compare the temperature differences (post–pre) among the three materials. Statistical analyses were conducted using SPSS version 22.0 (campus UEX license). A significance level of 5% (*p* < 0.05) was established.

## 3. Results

With the expanded polyethylene copolymer foam (material 1), significant increases in temperature were found after hours of use in all metatarsal heads, the styloid process, and the heel (Table 3, *p* < 0.001 in all cases). However, the temperature in the first toe did not increase after exercise (*p* = 0.170) (Table 3).

For the EVA foam (material 2), a significant increase in temperature was found in all regions of interest (ROIs) (Table 4, *p* < 0.001 in all cases). In particular, there was a notable increase in the areas of the fifth metatarsal head (pre, 24.5 °C ± 1.8 °C; post, 27.2 °C ± 2.2 °C), followed by the third (pre, 24.5 °C ± 1.8 °C; post, 27.0 °C ± 2.3 °C) and first metatarsal heads (pre, 24.4 °C ± 1.8 °C; post, 26.9 °C ± 2.4 °C), with increases of 2.7 °C in the fifth and 2.5 °C in the third and first metatarsal heads, respectively.

In the analysis of the expanded polyethylene–ethylene vinyl acetate copolymer foam (material 3), there was a significant increase in temperature in all regions of interest (Table 5). The plantar area of the fifth metatarsal head experienced the greatest temperature increase, with a difference of 2.2 °C between pre- and post-exercise measurements.

Comparing the temperature increments (post–pre) among the three materials, significant differences (*p* < 0.05 in all cases) were observed in the regions of interest (ROIs), except in the zone corresponding to the fifth metatarsal head, where there were no significant differences (*p* = 0.147). The data also indicate that material 2 (EVA) heats up more than material 1 (PE) and material 3 (PE-EVA). In material 2, the insole area that heats up the most is the one corresponding to the fifth metatarsal head, while the least heated area is the first toe (Table 6).

## 4. Discussion

Podiatrists, orthopedists, and other professionals who prescribe and fabricate foot orthotics need to understand the properties of materials to make decisions that align with the therapeutic needs of the patient [10]. Although absorption and heat and moisture transport properties determine the comfort of everything we wear or use to perform a function [11], these criteria had not been applied to the selection of materials for foot orthoses, which were chosen solely based on their mechanical characteristics (hardness, etc.) or the expected biomechanical effect (cushioning of vertical forces, etc.). The results of this study provide a tool for choosing between foot orthotic lining materials (PE, EVA, or PE-EVA), based on their thermal characteristics, thus providing the best option for foot orthosis users according to their needs.

PE is the material that exhibits the least overheating (1.17 °C) over its entire surface after 3 h of clinical activity, while EVA nearly doubles this temperature. This indicates that PE better dissipates the temperature generated inside the medical shoe, thus maintaining better comfort in the foot–material–shoe interface. Analyzing the results by zone, in all three materials, the first toe shows the lowest temperature increase (0.36–1.49 °C), while the fifth metatarsal head shows the greatest increase (1.85–2.73 °C). This could be because, during walking, these areas receive less pressure and friction (first toe) or more (fifth metatarsal head) [12], which is related to the increase in skin temperature and contact surface during walking [13].

Since the lining materials of orthoses are in direct contact with the sock/stocking or directly on the patient’s skin, they should have characteristics that benefit thermal comfort, such as high thermal conductivity, good air permeability, and a low friction coefficient, to facilitate heat transfer and thus reduce the temperature in the foot/insole/shoe microenvironment [14]. Therefore, commonly used materials such as EVA foam can cause an increase of 2.28 °C in temperature, which could lead to greater discomfort and sweating, potentially promoting the growth of fungi and bacteria in the long run [15].

The friction coefficient of these materials does not seem to have a direct relationship with the temperature increase because, although it might be expected that lower friction between the two surfaces (plantar skin and material) would result in less overheating, PE (medium friction coefficient) is the coolest material after activity, while EVA (low friction coefficient) shows the highest temperature increase. The hardness of the materials also does not seem to be related, as the hardness of PE or EVA is practically the same (25 vs. 27 Shore A), while EVA nearly doubles that of PE in temperature. The density of these three materials does appear to be a key factor in their thermal behavior, as they overheat in increasing order of density: EVA (2.28 °C, density 0.24 g/cm^3^), PE-EVA (1.43 °C, density 0.145 g/cm^3^), and PE (1.17 °C, density 0.11 g/cm^3^). The higher temperature in denser materials could be related, according to Lo et al. [16], to a lower moisture recovery rate. Thus, choosing a material with lower density could predict a lower overall temperature increase.

Another factor that could be related is the cellular structure of the material, as EVA is a material formulated with a closed-cell structure and is the one that heats up the most out of the three. The closed cell can trap more heat, while the open cell of PE and PE-EVA would allow for better temperature diffusion, keeping the material cooler. Currently, companies specializing in the distribution of materials for orthopedic insole manufacturing are researching new plant-based raw materials to prevent the temperature increase generated in users’ feet and with higher sustainability standards.

We believe that these results can shed light on the material selection process based not only on clinician preference or expected biomechanical effects but also on increased thermal comfort and therefore greater adherence to orthotic treatment. Thus, this thermal comfort depends mainly on controlling skin temperature and its reflection in the material in contact with it [17,18,19]. This factor would be particularly important in diabetic patients since the moisture absorption and thermal comfort of lining materials in these patients are important factors in ulcer prevention [20].

The lower overheating of PE may be the aspect that tips the choice of this material as the lining of foot orthoses in sports activities that generate greater friction on the metatarsal heads. However, it should be noted that all three materials efficiently regulate temperature, with discreet temperature increases, making all three suitable for regular daily activities. On the other hand, based on the results obtained, the use of EVA in activities that may generate greater friction due to frequent changes of direction and in people with sweating problems or prone to skin injuries should be reconsidered and avoided, as it is the material that generates the highest temperature increase after use. 

These considerations could be very useful for the design and implementation of future intelligent plantar orthoses, in which the use of EVA should be avoided due to it being the material that registers the highest temperature in the study, and priority should be given to PE and PE-EVA materials due to their more moderate thermal behavior. In the future, intelligent elements could be developed in the form of insoles to record health and comfort data; thus, it would be advisable to take into account the results of thermal studies for their greater effectiveness.

Therefore, in the future, technology applied to the commercialization of intelligent insoles must take into account the different thermal behaviors of the materials used in their manufacture to properly enable their use in daily activities and thus minimize potential errors. These results must be taken into account because the design of sensors integrated into orthopedic insoles to monitor foot health in real time might integrate the thermal characteristics of materials to avoid bias in the measurements. This issue is extremely important in wearable insoles for diabetic patients, thus requiring accurate temperature measurement [21,22].

As for the present study, there are limitations that may have affected the accuracy of the results. The first is that the study was conducted without socks or stockings to avoid the influence of different compositions on the temperature generated by the participants. Additionally, the short duration of 3 h of activity prevents the analysis of the real impact of the lining materials on thermal regulation and their relationship with comfort.

## 5. Conclusions

Polyethylene foam’s resistance to overheating makes it an ideal lining material for foot orthoses, with it being especially noted for its effective thermal management in critical areas. The area around the hallux remains notably cooler, whereas the fifth metatarsal head tends to accumulate more heat. The foam’s physical and chemical properties, such as its density and cellular structure, significantly influence its ability to regulate temperature; materials that are less dense and feature open cells are particularly effective at minimizing temperature rises. In contrast, the use of EVA (ethylene vinyl acetate) should be avoided in situations involving high friction at the metatarsal heads, like activities requiring frequent directional changes, or in individuals prone to excessive sweating or skin lesions, as EVA tends to retain more heat, exacerbating these issues.

## Figures and Tables

**Figure 1 sensors-24-02928-f001:**
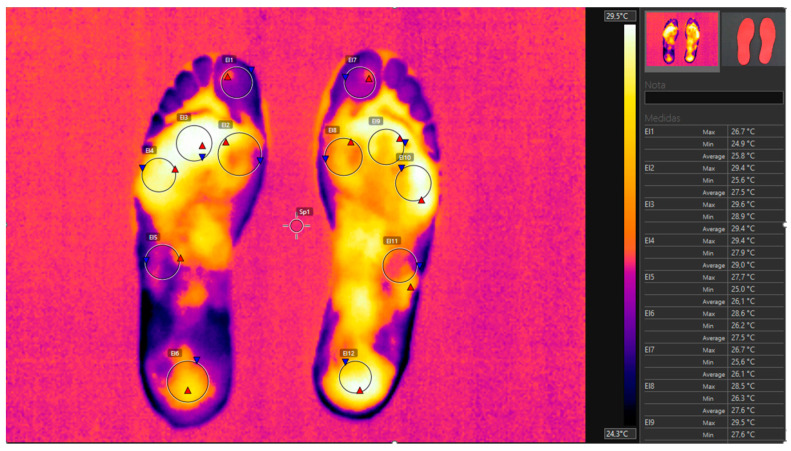
Thermal image of the insole after 3 h of use.

**Table 1 sensors-24-02928-t001:** Descriptive statistics of the participants.

	Minimum	Maximum	Mean	Standard Deviation (SD)
Age	19	51	24.6	8.2
Foot size (EU)	35	46	40.3	2.9
Mass (kg)	47	105	67.1	13.6
Height (m)	1.5	120.9	1.6	0.09

**Table 2 sensors-24-02928-t002:** Comparison of anthropometric characteristics between men and women.

	Gender	Mean	Standard Deviation (SD)	*p*
Age	Men	26.9	12.0	0.160
	Women	23.0	3.0	
Foot size	Men	43.4	1.5	<0.001
	Women	38.2	1.4	
Weight	Men	75.0	13.3	0.002
	Women	61.5	10.9	
Height	Men	1.7	0.08	<0.001
	Women	1.6	0.06	

**Table 3 sensors-24-02928-t003:** Paired *t*-test statistics on insoles of material 1, polyethylene copolymer foam. MTHL: metatarsal head.

ROIs		Mean	Standard Deviation (SD)	*p*
1st Toe	Pre	23.7	1.9	0.170
	Post	24.1	2.2	
1st MTH	Pre	23.7	1.9	<0.001
	Post	24.1	2.2	
3rd MTH	Pre	23.8	1.9	<0.001
	Post	25.2	2.1	
5th MTH	Pre	23.8	1.9	<0.001
	Post	25.6	2.1	
Styloid Process	Pre	23.8	1.9	<0.001
	Post	24.9	1.9	
Heel	Pre	23.9	1.9	<0.001
	Post	24.9	2.0	

**Table 4 sensors-24-02928-t004:** Paired *t*-test statistics on insoles of material 2, EVA; MTH: metatarsal head.

ROIs		Mean	Standard Deviation (SD)	*p*
1st Toe	Pre	24.5	1.8	<0.001
	Post	26.0	2.6	
1st MTH	Pre	24.4	1.8	<0.001
	Post	26.9	2.4	
3rd MTH	Pre	24.5	1.8	<0.001
	Post	27.0	2.3	
5th MTH	Pre	24.5	1.8	<0.001
	Post	27.2	2.2	
Styloid Process	Pre	24.5	1.8	<0.001
	Post	26.6	2.4	
Heel	Pre	24.6	1.9	<0.001
	Post	26.9	2.3	

**Table 5 sensors-24-02928-t005:** Paired *t*-test statistics on insoles of material 3, polyethylene–ethylene vinyl acetate (PE-EVA) copolymer. MTH: metatarsal head.

ROIs		Mean	Standard Deviation (SD)	*p*
1st Toe	Pre	24.0	2.0	0.040
	Post	24.6	2.2	
1st MTH	Pre	24.0	2.0	<0.001
	Post	25.7	2.3	
3rd MTH	Pre	24.0	2.0	<0.001
	Post	25.8	2.2	
5th MTH	Pre	24.0	2.0	<0.001
	Post	26.2	2.0	
Styloid Process	Pre	24.1	1.9	<0.001
	Post	25.6	2.2	
Heel	Pre	24.1	2.0	0.002
	Pre	24.1	2.0	

**Table 6 sensors-24-02928-t006:** Comparison of temperature increments among the three materials.

ROI’S	M1	M2	M3	W Mauchly (Sig)	Pillai’s Trace	*p*
	Mean Increase °C					
Hallux	0.36	1.49	0.58	0.932 (*p* = 0.302)	0.213	0.017
1st MTH	1.31	2.43	1.64	0.959 (*p* = 0.488)	0.165	0.047
3rd MTH	1.40	2.54	1.77	0.948 (*p* = 0.406)	0.163	0.049
5th MTH	1.85	2.73	2.16	0.958 (*p* = 0.479)	0.107	0.147
Styloid Process	1.07	2.14	1.48	0.986 (*p* = 0.787)	0.184	0.031
Heel	1.04	2.35	1.00	0.923 (*p* = 0.255)	0.284	0.003
Mean	1.17	2.28	1.43			

## Data Availability

Data are available at http://www.unex.es/investigacion/grupos/biopiex (accessed on 19 January 2024).

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
