# Peer review of "Analyzing the Thermal Characteristics of Three Lining Materials for Plantar Orthotics"

_sensors, 2024, doi:10.3390/s24092928_

Round 1

Reviewer 1 Report

Comments and Suggestions for Authors

I have read this paper with interest. The study addresses an important aspect: the thermal behavior of lining materials used in orthotics. It goes beyond traditional criteria (hardness, breathability) and focuses on temperature response. While material properties have been studied, this research specifically examines thermal characteristics of three common materials used as liners in orthotic therapy. 

The methods section is excellently presented, providing precise details on the protocol utilized for this study. This meticulous approach not only ensures clarity but also helps in minimizing bias.

The results section effectively presents the findings of the study, offering clear and concise summaries of the data collected. The discussion section provides insightful interpretations of the results, offering valuable insights into the implications and significance within the context of the research objectives. Implications of findings were also discussed in the context of emerging new technologies and smart insoles.

Comments on the Quality of English Language

The English used was of a good level. There are some minor typos which need to be addressed by the authors. 

Author Response

Thank you very much for your opinion about our work.

The corrections minor typos mistakes has been corrected, being undelined in yellow in the text.

Regards

Reviewer 2 Report

Comments and Suggestions for Authors

This article hypothesises that some intrinsic properties, such as the type of material, the density or friction coefficient of friction, can influence the heating of the material, which can have a negative impact on the perception of comfort. The aim of this study was therefore to evaluate the temperature rise of the three types of foam commonly used as insoles after a period of use in a clinical environment.

The article is well written and interesting to read. However, I see the following minor issues that should be resolved before publishing this article:

The introduction should refer to other authors working on this topic. It is necessary to describe how these studies differ from similar studies and what the scientific contribution is.

18: “Weight" should be replaced by "mass”. Correct “kg” with “kg”.

Table 1: Correct “kg” with “kg” (unpr. OK 75)

Tables 1 and 2 show SD, tables 3-5 standard deviation (SD). It must be the same

Density g/m3 to be corrected by 3 in the exponent.

The conclusion is very poor and must be supplemented.

Author Response

Dear reviewer 2. Thank you for your very helpful revision of our work, we apreciate your opinion. The changes due to your work are marked in the text in green.

The introduction should refer to other authors working on this topic. It is necessary to describe how these studies differ from similar studies and what the scientific contribution is.

Response: Additional info has been added in the introduction.

18: “Weight" should be replaced by "mass”. Correct “kg” with “kg”.

Response: These changes are been corrected in the text

Table 1: Correct “kg” with “kg” (unpr. OK 75)

Response: This change are been corrected in the text

Tables 1 and 2 show SD, tables 3-5 standard deviation (SD). It must be the same

Response: These changes has been corrected in the text

Density g/m3 to be corrected by 3 in the exponent.

Response: These changes has been corrected in the text

The conclusion is very poor and must be supplemented.

Response: The conclusion section has been supplemented.

Reviewer 3 Report

Comments and Suggestions for Authors

Results of a methodologically well organized research are presented in the manuscript. The manuscript is well designed and arranged, all data are presented in a logical sequence, discussion part is sufficient. I would mention only one critical recommendation for future investigations: Thermal characteristics of liner materials for plantar orthotics must always be analysed together with humidity absorption/water vapour permeability (especially for diabet foot).

Also there are some editorial remarks that must be corrected before article' publishing:

- g/cm3 must be corrected to g/cm3 (Lines 97,101,106, 236, 237)

- integers must be separated by a dot, not a comma (Tables 5 and 6)

- the same sentences in Lines 160 - 167 are repeated twice.

Author Response

Dear Reviewer 3. Thank you for your words about our work. Your suggestions are very helpful to improve the paper.

Thank you to recommed that humidity absorption/water vapour permeability must be measured in further studies. We will do it in this way in future works.

The changes in the text due to your work has been marked in blue in the text.

g/cm3 must be corrected to g/cm3 (Lines 97,101,106, 236, 237)

Response: these changes has been added to the text.

Integers must be separated by a dot, not a comma (Tables 5 and 6)

Response: These changes has beed added to the text.

The same sentences in Lines 160 - 167 are repeated twice.

Response: This change has been implemented in the text. Second time was removed of the text.